# Role of microRNAs in Immune Regulation with Translational and Clinical Applications

**DOI:** 10.3390/ijms25031942

**Published:** 2024-02-05

**Authors:** Zsuzsanna Gaál

**Affiliations:** Department of Pediatrics, Faculty of Medicine, University of Debrecen, 98 Nagyerdei krt, 4032 Debrecen, Hungary; zsuzsanna.gaal.46@gmail.com

**Keywords:** microRNA, autoimmunity, anticancer immunity, transplantation immunology, immunomodulation, epigenetics, biomarkers, personalized treatment

## Abstract

MicroRNAs (miRNAs) are 19–23 nucleotide long, evolutionarily conserved noncoding RNA molecules that regulate gene expression at the post-transcriptional level. In this review, involvement of miRNAs is summarized in the differentiation and function of immune cells, in anti-infective immune responses, immunodeficiencies and autoimmune diseases. Roles of miRNAs in anticancer immunity and in the transplantation of solid organs and hematopoietic stem cells are also discussed. Major focus is put on the translational clinical applications of miRNAs, including the establishment of noninvasive biomarkers for differential diagnosis and prediction of prognosis. Patient selection and response prediction to biological therapy is one of the most promising fields of application. Replacement or inhibition of miRNAs has enormous therapeutic potential, with constantly expanding possibilities. Although important challenges still await solutions, evaluation of miRNA fingerprints may contribute to an increasingly personalized management of immune dysregulation with a remarkable reduction in toxicity and treatment side effects. More detailed knowledge of the molecular effects of physical exercise and nutrition on the immune system may facilitate self-tailored lifestyle recommendations and advances in prevention.

## 1. Introduction

MicroRNAs (miRNAs) are about 19–23 nucleotide long, evolutionarily conserved noncoding RNA molecules that regulate gene expression at the post-transcriptional level by targeting the 3′-untranslated region (UTR) of specific messenger RNAs (mRNAs) [1]. miRNAs are generally transcribed by RNA polymerase II [2] and processed through a series of steps in the nucleus and afterwards in the cytoplasm [3]. Primarily transcribed pri-miRNAs are processed to single hairpins termed precursor miRNAs (pre-miRNAs) by a double-stranded RNA (dsRNA)-specific endonuclease enzyme Drosha and the dsRNA-binding protein DiGeorge Critical Region 8 (DGCR8). Pre-miRNAs are shuttled to the cytoplasm by the nuclear export factor Exportin 5 (Exp5) and in the cytoplasm further processed into functional 21–23-nt RNAs by the RNase III-type endoribonuclease enzyme Dicer [4,5]. Processed miRNA duplexes are incorporated into the RNA-induced silencing complex (RISC) that is directed by its bound small RNA to target complementary mRNAs and repress their translation through mRNA cleavage or degradation [6]. Recently described modifications during miRNA processing include alterations of N6-methyladenosine (m6A) and 7-methylguanosine (m7G), adenosine-to-inosine (A-to-I) editing and pseudourylation (ψ) [5]. DNA-methylation and histone modifications also affect cotranscriptional and/or post-transcriptional pri-miRNA processing [7].

Since their initial discovery in 1993 by Lee and colleagues in *Caenorhabditis elegans* [8], approximately 2600 mature microRNAs (miRBase v.22) were identified in the human genome and more than 200,000 of miRNA transcripts (GENCODE v.29) [9]. miRNAs are powerful regulators of various cellular activities including differentiation, cell growth and apoptosis [10]. Dysregulation and abnormal expression of miRNAs have been linked to multiple human diseases, including metabolic disorders, cancer and autoimmune diseases [11].

miRNAs were initially associated with the regulation of immune responses by a report in 2004 showing selective expression of miR-181a and miR-223 in B-lymphocytes and myeloid cells, respectively [12]. Since then, miRNAs have been confirmed to play a pivotal role in both innate and acquired immunity. Furthermore, the development of numerous immunological disorders are directly attributed to dysregulated miRNA expression [13]. Clinical applications of miRNAs as biomarkers and therapeutic targets are facilitated by their widespread abundance in body fluids and resistance to degradation over long-term storage [13,14].

In this review, involvement of miRNAs is summarized in the differentiation of immune cells, in anti-infective immune responses, immunodeficiencies and autoimmune diseases. Roles of miRNAs in anticancer immunity and in the transplantation of solid organs and hematopoietic stem cells are also discussed (Figure 1). Major focus is put on their translational and clinical applications, highlighting the emerging possibilities for personalized management of immunological diseases by advances in prevention and novel therapeutic interventions. However, utilization of miRNA fingerprints as biomarkers for diagnosis, prognosis and prediction of treatment response requires solutions for important technical challenges. Deeper insights into the influence of nonpharmacological immunomodulation on miRNA patterns links lifestyle-related molecular changes to transgenerational epigenetic inheritance.

## 2. miRNAs Regulate the Differentiation and Function of Immune Cells

miRNAs are implicated in the fine-tuning of differentiation and regulation of immune cells. Disruptions of miRNA biogenesis machinery and deregulation of miRNA expression result in impaired function of both innate and adaptive immunity.

### 2.1. T Lymphocyte

Numerous miRNAs have been confirmed to regulate the differentiation of T-cells into diverse types of effector cells [15], such as miR-410 being implicated in the fine-tuning of the differentiation of helper T-cells (Th) via the PI3K-AKT-VEGF-signaling pathway [16]. In murine models, the deficiency of miR-21 resulted in Th1 prone animals [17], whereas the overexpression of miR-138 increased the frequency of Th2 cells by increased levels of IFNγ and decreased expression of RUNX3 transcription factor, respectively [18]. Up-regulation of miR-19a was observed in both human and murine Th2 cells, that promoted the differentiation of Th2 cells by targeting phosphatase and tensin homolog (*PTEN*) and suppressor of cytokine signaling 1 (*SOCS1*) genes [19]. Overexpression of miR-34a in human CD4^+^ and CD8^+^ T-cells led to a significant decrement of nuclear factor-kappa beta (NFκB) inhibitor alpha (NFΚBIA) and reduced cell surface abundance of T-cell receptor (TCR) subunits, resulting in impaired T-cell killing capacity [20]. Differentiation and function of CD8^+^ cytotoxic T-cells is regulated by miR-143 [21], while increased miR-146a and decreased miR-21 levels significantly enhanced the immune-suppression triggered by regulatory T-cells (Tregs) [22]. Deficiency of miR-15/16 results in the accumulation of functionally impaired Tregs, loss of immune tolerance and the development of spontaneous multi-organ inflammation in a mouse model [23]. miR-181a has been shown to be an important regulator of the activation threshold of the TCR [24].

### 2.2. Natural Killer Cell

Several miRNAs have been identified to regulate the development of natural killer (NK) cells including miR-15/16, miR-155, miR-181, miR-483-1 and miR-583 [25]. miR-15/16 controls the NK cell maturation program by directly targeting c-Myb transcription factor, that is expressed preferentially by immature NK cells [26]. Genetic deletion of miR-150 in mice led to a significant reduction in the number of mature NK cells [27]. In human NK cells, overexpression of miR-155 was strongly associated with enhanced production of gamma interferon (IFNγ) in response to cytokine stimulation [28]. miR-27a* was reported to affect the function of NK cells by targeting their cytotoxic effector molecules perforin-1 and granzyme-B [29].

### 2.3. B Lymphocyte

The role of microRNAs has been confirmed in both B-cell differentiation and B-cell tolerance [30]. miR-150 was found to be a critical regulator of proliferation, differentiation, metabolism and apoptosis of B-cells, and altered expression of which was established as a diagnostic biomarker of various autoimmune diseases [31]. miR-130b is involved in the molecular regulation of B-cell activation through modulating the NFκB pathway by targeting the lysine deubiquitinase CYLD gene [32]. miR-155-mediated repression of the histone lysine demethylase enzyme KDM2A is essential to prevent the excessive production of reactive oxygen species (ROS) and apoptosis in high-affinity clones of positively selected germinal center (GC) B-cells, thereby ensuring subsequent affinity maturation [33]. Deficiency of the Dicer enzyme in B-cells resulted in a skewed repertoire of B-cell receptor (BCR), hallmarks of autoreactivity and high-serum titers of autoreactive antibodies in female mice [30].

### 2.4. Macrophage

miRNAs fine-tune the development and functions of macrophages including phagocytosis, efferocytosis and tissue repair. Moreover, microvesicles (MVs) released by macrophages have an essential role in intercellular communication by carrying functionally active miRNAs [34]. The maturation of macrophages is mainly controlled by miR-15, miR-146a, miR-338 and miR-342 [35], while different forms of macrophage activation are featured by specific miRNA patterns. miR-155, miR-181a and miR-451 were up-regulated and miR-125-5p, miR-143-3p and miR-146a-3p were down-regulated in classically activated macrophages (M1) compared to the alternative form (M2) [36]. M1 macrophage polarization was attenuated by miR-223 via suppressing the Notch signaling pathway and Nod-like receptor protein 3 (NLRP3)-mediated pyroptosis in experimental autoimmune uveitis (EAU) [37]. On the other hand, M2 polarization of macrophages was facilitated by miR-214 via targeting of the glycogen synthase kinase 3 beta (GSK3B) enzyme in a model of allergic rhinitis [38]. Overexpression of miR-181c-5p in bone marrow-derived macrophages resulted in increased phagocytic ability through the regulation of cytoskeletal arrangement [39].

### 2.5. Dendritic Cell

Dendritic cells (DC) have a unique role within the immune system, linking innate and adaptive immune responses [40]. Transition of DCs from steady state to mature state is facilitated by miR-9 in a cell-type-specific manner [41]. miR-150 and miR-223 are involved in the regulation of antigen presentation ability, while miR-29c, miR-146 and miR-155 affect the survival of DCs [42]. Up-regulation of miR-338 in DCs led to pathogenic Th17 responses and exacerbated the development of EAU in an animal model [43]. Overexpression of miR-200b in human peripheral blood mononuclear cells (PBMCs) resulted in significantly reduced numbers of protruding veils in mature DCs that are critical for antigen-specific T-cell activation [44].

### 2.6. Neutrophil

miRNAs play a fundamental role in both differentiation and function of neutrophil granulocytes. While the elevated expression level of miR-155 (targeting PU.1 transcription factor) features myeloblasts and promyelocytes, high levels of miR-27a (down-regulating RUNX1 transcription factor) are characteristic to myelocyte stages, and miR-223 is the most well-known miRNA that facilitates the process of granulopoiesis [45]. Regulation of neutrophil actin cytoskeleton by miR-142 is essential for protection against bacterial infections at skin wound sites [46]. Based on the results of experimental inhibition, miR-328 and miR-183/96/182 clusters influence the phagocytic capacity of neutrophil granulocytes [47,48], while zebrafish miR-722 was identified as a suppressor of chemotaxis in human neutrophil-like cells [49].

### 2.7. Mast Cell

Mast cells are key regulators of allergic disorders and immediate hypersensitivity reactions [50]. Compared to other hematopoietic lineages, bone marrow-derived mast cells expressed lower levels of miR-223higher levels of miR-24, miR-26a and miR-27a [51]. Dinitrophenol (DNP)-stimulated and resting mast cells also showed different miRNA fingerprints including the expression of let-7i, that was confirmed to inhibit the process of degranulation by targeting exocyst complex component 8 (Exco8) [52]. Degranulation is also regulated by miR-21 through the inhibition of p38 pathway [53]. According to recently published data, exosomal miRNAs derived from mast cells are implicated in the development of tumor metastasis, such as the transfer of miR-490 to hepatocellular carcinoma cells resulted in the inhibition of metastasis formation through the regulation of EGFR/AKT/ERK pathway [54].

## 3. Role of miRNAs in Anti-Infectious Immunity

Post-transcriptional regulation of gene expression by miRNAs has a fundamental role in promoting and fine-tuning the appropriate immune response in case of infections [55] (Table 1). miRNAs can be applied as diagnostic biomarkers and therapeutic targets in a wide variety of infectious diseases [56].

### 3.1. Antiviral Immunity

miRNAs have been confirmed to influence both the replication of viruses and the antiviral immune response of the host. Intact regulation of miR-155 was confirmed to have key significance in the development of an effective antiviral immune response without any harmful consequences regarding the host: miR-155 regulates interferon response, macrophage polarization, activity of NK cells and the generation of antiviral antibodies as well [85].

Replication of hepatitis B virus (HBV) is suppressed by miR-1231 by targeting HBV core mRNA [57]. In contrast, hepatitis C virus (HCV) replication is facilitated by miR-122, that can be therapeutically targeted by a locked nucleic acid (LNA)-modified oligonucleotide (SPC3649/miravirsen) complementary to miR-122, leading to long-lasting suppression of viraemia [58].

Recovery from viral pneumonia is featured by a complex interplay between expression of specific miRNAs, lung regenerationinflammation. Inhibition of miR-21 and miR-99a resulted in deleterious molecular and cellular effects on pulmonary repair and inflammatory processes during influenza pneumonia in proliferating alveolar type II (AT2) cells in a mouse model [59]. IFN production in human metapneumovirus (HMPV)-infected macrophages is regulated by miR-4634 [60]. Synthesis of a stimulator of interferon genes (STING) was found to be attenuated by miR-24, a miRNA that is induced by herpes simplex virus 1 (HSV1) in order to evade cellular antiviral response [61].

High levels of miR-BART17-3p in Epstein–Barr virus (EBV)-infected NK cells down-regulated the retinoic acid-inducible gene I (RIG-I)-like receptor antiviral pathway, leading to enhanced expression of EBV-encoded proteins [62]. Deregulation of host miRNAs also affect pathways related to antiviral immune response. Down-regulation of miR-194, miR-302b and miR-302c, whereas up-regulation of miR-30c in Zika virus (ZIKV)-infected human neuronal precursor cells contribute to the development of microcephaly and brain lesions [63]. 

Replication of human immunodeficiency virus 1 (HIV-1) is inhibited by miR-181-5p in lymphocytes by the down-regulation of dead-box protein 3 (DDX3X), a host factor in HIV-1 nuclear export [64]. Since it negatively regulates Toll-like receptor (TLR)-mediated cytokines, targeting miR-155 might be a novel strategy towards alleviating HIV disease progression [86].

According to recently published data, miRNA fingerprints can also be applied as prognostic biomarkers and novel therapeutic targets in SARS-CoV-2 infection. One of the most recent studies in the field identified let-7, miR-155 and miR-223 as potential biomarkers for distinguishing healthy individuals from COVID-19 patients [87]. The prediction of mortality was supported by low levels of miR-1-3p, miR-23b-3p, miR-141-3p, miR-155-5p and miR-4433b-5p with high specificity and sensitivity, while the combined evaluation of serum miR-21-5p, miR-22-3p, miR-155-5p and miR-224-5p could discriminate mild/moderate COVID-19 patients from severe cases [88]. In COVID-19 Egyptian patients, decreased expression level of miR-18a-5p was identified as a bad prognostic marker [65]. Recent evidence suggests the pathogenetic role of deregulated pro- and antifibrotic miRNAs (including miR-17-5p, miR-19a-3p, miR-20a-5p and miR-200c-3p) in the development of fibrotic lung lesions in post-COVID patients [67]. Obefazimod (ABX464) is a small molecule that up-regulates miR-124 in immune cells that reduces the production of inflammatory cytokines and prevents the proliferation of Th17 cells. However, once-daily oral obefazimod in a dose of 50 mg provided no benefit over placebo for severe COVID-19 in a randomized, double-blind placebo-controlled trial (miR-AGE) [66].

### 3.2. Immune Response against Bacteria

miRNAs are involved in the regulation of host immune defenses against bacteria, for example, the up-regulation of miR-30e-5p was shown to enhance signaling pathways of innate immunity in numerous bacterial infections by targeting key negative regulators such as SOCS1 and SOCS3 [89]. miR-155, one of the most crucial miRNAs in shaping the host immune defenses against *Mycobacterium tuberculosis*, was found to be down-regulated in patients with clear clinical symptoms of tuberculosis in comparison with healthy individuals and latently infected patients [68]. Up-regulation of miR-215-5p was identified as a biomarker of pulmonary tuberculosis, that prevents the fusion of autophagosomes with lysosomes in macrophages [69]. miR-150 suppressed apoptosis of bovine monocyte-derived macrophages infected with *Mycobacterium avium* subsp. *paratuberculosis* by targeting programmed cell death protein-4 (PDCD4) [70].

In pneumococcal pneumonia patients, down-regulation of miR-223 in serum correlated with increased disease severity [71]. Induction of endotoxin tolerance in case of meningococcal infection was found to be facilitated by the up-regulation of miR-146a [72]. *Legionella pneumophila* infection caused a pro-inflammatory reaction in murine macrophages through the induction of miR-125a-3p, that targets N-terminal asparagine amidohydrolase 1 (NTAN1) enzyme [73]. Lung injury induced by *Klebsiella pneumoniae* infection was alleviated by miR-181a-5p, delivered by adipose-derived mesenchymal stem cell (ADSC)-released exosomes [74]. Intracellular survival of *Brucella abortus* was found to be suppressed by miR-125b-5p through targeting the cytoplasmic zinc finger protein A20, resulting in the activation of NFκB signaling and increased production of tumor necrosis factor α (TNFα) [75].

### 3.3. miRNA Fingerprints in Fungal Infections

*Candida albicans*-induced acute lung injury was suppressed by the down-regulation of miR-155 via activating SOCS1 in a murine model [76], while invasive *Candida albicans* infection was accompanied by a remarkable increment of miR-16-1 and decrement of miR-17-3p expression in respiratory epithelial tissues of infected individuals [77]. Down-regulation of miR-146a was confirmed in *Candida glabrata*-infected monocyte-derived macrophages, that may contribute to the reduction in pro-inflammatory cytokine production [78]. Vast majority of the identified 33 miRNAs that regulate genes related to the pathogenesis of invasive aspergillosis (such as *S100B*, *TDRD9* and *TMTC1*) were also linked to the fine-tuning of platelet activation [90]. Overexpression of miR-21-5p, miR-26-5p, miR-142-3p and miR-142-5p in cases of invasive aspergillosis in hemato-oncology patients with profound neutropenia may be applied as a promising diagnostic adjunct with sufficient accuracy and sensitivity [79].

### 3.4. Antiparasite Immunity

Dysregulation of miRNAs contributes to dampened immune responses and, moreover, altered expression patterns are novel indicators of organ manifestations in parasitic infections. Overexpression of bma-miR-34 by *Brugia malayi/microfilariae* impairs the migration and activation of immune cells by the inhibition of CXCL10/CXCL11/CXCR3 secretion [80]. Lack of miR-155 resulted in a significant reduction in CD8-positive and NK cells, leading to diminished survival of mice infected with *Trypanosoma cruzi* [81]. *Leishmania donovani*-infected monocyte-derived dendritic cells are featured by the dysregulation of miR-21 and miR-146b-5p [82]. Chronic soil-transmitted helminth infections are featured by the down-regulation of let-7d that contributes to a modified Th2 immune response [91].

In a murine model, alterations of mmu-miR-146b and mmu-miR-155 expression in the mid-phase (day 30) and late phase (day 45) of *Schistosoma japonicum* infection were related to the regulation of hepatic inflammatory response and the development of hepatopathy [83]. In contrast to noncerebral malaria, increased levels of miR-27a and miR-150let-7i were confirmed in mice with cerebral malaria [84]. 

### 3.5. miRNA Biomarkers in Sepsis

Altered miRNA fingerprints are implicated in both the pathogenesis and early detection of septic conditions. Enhanced expression of chemokine ligand 2 (CCL-2) in macrophages, induced by miR-155 mimics, subsequently enhanced lung injury in a lipopolysaccharide (LPS)-induced murine model for endotoxemia [92]. miR-125b-5p, delivered by exosomes from bone marrow-derived mesenchymal stem cells (MSCs) of adult C57BL/6J mice, inhibited pyroptosis of macrophages and alleviated sepsis-associated acute lung injury via the down-regulation of the signal-transducer and activator of the transcription 3 (STAT3) protein [93]. miR-494-3p was found to protect rat cardiomyocytes against septic shock through the regulation of PTEN expression [94].

According to recently published results, a three-miRNA panel (miR-15a-5p, miR-16-5p and miR-223-3p) may be applied as a noninvasive marker for early onset sepsis of neonates [95], whereas miR-519c-5p and miR-3622b-3p were identified as novel biomarkers of sepsis in adult patients [96]. Furthermore, evaluation of serum miR-155 expression can be utilized as a novel marker for septic acute kidney injury (AKI) [97].

## 4. Implications of miRNAs in Immunodeficiencies 

Growing numbers of miRNAs are associated with the pathogenesis of both primary immunodeficiencies and disturbances of immune regulation, that may serve as novel therapeutic targets as well. 

In patients with common variable immunodeficiency (CVID) without identified genetic defects, the terminal differentiation of B-cells was found to be affected by the overexpression of miR-125b-5p, via the down-regulation of B-lymphocyte-induced maturation protein 1 (BLIMP-1) and interferon regulatory factor 4 (IRF-4) transcription factors [98]. High levels of miR-210 are suggested to have a role in the development of autoimmunity in CVID patients [99], while miR-142 and miR-155 have been implicated in neoplastic clinical complications of CVID including gastric mucosa-associated lymphoid tissue (MALT) lymphoma and natural killer/Tcell lymphoma (NKTCL) [100]. Up-regulation of miR-33 in alveolar macrophages by pro-inflammatory cytokines may perpetuate chronic inflammatory granulomatous disease by repressing ATP-binding cassette (ABC) lipid transporters ABCA1 and ABCG1 [101]. Dysregulation of the autoimmune regulator (AIRE) gene by miR-220b may be a causal factor for autoimmune polyendocrinopathy-candidiasis-ectodermal dystrophy (APECED) or its related diseases [102].

Up-regulation of miR-30e diminished NK cell cytotoxicity in patients with hemophagocytic lymphohistiocytosis (HLH) by targeting granzyme B and perforin [103]. Plasma miR-BART16-1, encoded by the BamHI-A region rightward transcript (BART) of EBV, could be a potential biomarker for monitoring the progression of EBV-induced HLH [104].

## 5. miRNAs as Potential Biomarkers and Therapeutic Targets in Autoimmune Diseases

Dysregulation of miRNAs have been confirmed in autoimmune diseases, providing a wide range of clinical applications. In this section, examples for the involvement of miRNAs in disease pathogenesis are followed by their potential applications as diagnostic biomarkers, prognostic tools and novel therapeutic targets (Table 2).

### 5.1. Rheumatoid Arthritis

Besides global down-regulation of their miRNA contents, neutrophil granulocytes derived from patients with rheumatoid arthritis (RA) also have been confirmed to have a defect in the machinery of miRNA biogenesis [157]. Polymorphisms in miR-146a (rs2431697) and in its target interleukin-1 receptor-associated kinase 1 (IRAK1) (rs3027898) have been linked to increased susceptibility for RA in Chinese and Greek populations, respectively [105,158]. Migration and angiogenesis of human umbilical vein endothelial cells (HUVEC) was facilitated by TNFα-induced exosomes derived from fibroblast-like synoviocytes of RA patients via the miR-200a-3p/KLF6/VEGFA axis [106].

Levels of miR-371b, miR-483 and miR-642b were found to be significantly up-regulated, whereas miR-25 and miR-378d were down-regulated in PBMCs of patients who developed RA from early undifferentiated arthritis [108]. Expression level of miR-21-5p positively correlated with Disease Activity Score 28-joint count with erythrocyte sedimentation rate (DAS28-ESR) and tenosynovitis in gray-scale ultrasound (GSUS), whereas negative correlation was confirmed with the absolute counts of Tregs [107]. Expression levels of circulating miR-23 and miR-223 can be applied as predictors of response to anti-TNFα therapy [110], while circulating miR-19b is a promising novel biomarker of treatment response to Janus kinase (JAK)-inhibitor baricitinib in RA patients [109].

Since its deficiency resulted in impaired polarization of Th17 cells and prevented the generation of autoreactive T- and B-cells in a murine model of autoimmune arthritis, miR-155 might provide a novel therapeutic target for the treatment of RA [111]. miR-340-5p was confirmed to suppress the proliferation of fibroblast-like synoviocytes (FLS) by targeting STAT3 transcription factor, therefore, its induction may also serve as a potential target for the treatment of RA [112].

### 5.2. Systemic Lupus Erythematosus

Besides the growing numbers of genes and phenotype-related loci that are involved in the clinical heterogeneity of systemic lupus erythematosus (SLE), epigenetic alterations are also considered to have an association with clinical features of the disease [159]. Polymorphisms within the host gene of miR-17-92 (rs4284505) and in a region that encodes a miR-146-regulator immune cell specific enhancer (rs2431697) are associated with increased susceptibility to SLE [113,160]. Expression of miR-146a was found to be significantly down-regulated in SLE patients, contributing to the activation of type I interferon pathway [114], while the modulatory effects of miR-365a-3p on the IL6 target gene [115] and down-regulation of the miR-4689 sponge circular RNA circPTPN22 was also implicated in the pathogenesis of the disease [116].

Evaluating the expression of circulating miR-124-3p, miR-320b and miR-377-3p levels may constitute promising biomarkers for the diagnosis of SLE [118,119]. In the clinically active group of patients, significant up-regulation of serum miR-21 was detected, and a significant positive correlation has been established between the expression of miR-146 and disease activity according to the SLE Disease Activity Index 2000 (SLEDAI-2K) [117]. Higher stages of lupus nephritis (LN) were associated with up-regulation of miR-181a and down-regulation of miR-223 expression, that can be applied in the evaluation of disease progression [120]. Serum miR-485-5p might have a predictive value for the risk of end-stage renal disease (ESRD) in LN patients [121].

Administration of a miR-7 antagomiR resulted in the correction of PTEN-related hyperresponsiveness of B-cells derived from newly diagnosed untreated SLE patients [122]. According to recent results in a murine model, miR-590-3p mimics inhibited Th17 cells by suppressing autophagy, that may improve clinical symptoms of lupus [125]. Overexpression of miR-30a was found to ameliorate podocyte injury in LN through the suppression of Notch1 signaling pathway [123]. Similarly, up-regulation of miR-181d-5p also has therapeutic potential to improve LN by its impact on the mitogen-activated protein kinase 8 (MAPK8) enzyme [124].

### 5.3. Antiphospholipid Syndrome

Expression levels of miR-19b-3p and miR-20a-5p in monocytes derived from patients with antiphospholipid syndrome (APS) were found to be low, especially in cases of high levels of antiphospholipid antibodies [127]. Exosomal miR-146a-5p derived from human umbilical cord MSCs can alleviate trophoblast injury and placental dysfunction by regulating the tumor necrosis factor receptor-associated factor 6 (TRAF6)/NFκB axis [128]. Small extracellular vesicles (sEVs) containing miRNAs are involved in the pathogenesis of APS, such as the up-regulation of miR-483-3p and down-regulation of miR-326 through the regulation of phosphatidylinositol-3-kinase (PI3K)-AKT and Notch pathways, respectively [126].

### 5.4. Scleroderma and Systemic Sclerosis

Increased expression of miR-21 has been confirmed to contribute to the uncontrolled fibrosis in systemic sclerosis (SSc) through the regulation of transforming growth factor β (TGFβ) signaling [129], while the level of miR-29a correlated with both that of miR-21 and the presence of anti-Scl70 antibodies in the serum [132]. Epigenetic down-regulation of miR-126 in scleroderma endothelial cells was associated with defective angiogenesis, that could be successfully restored by azacitidine and trichostatin A treatment [130]. Th17 cell-derived miR-155-5p is also considered to be an important factor of SSc pathogenesis by its regulatory role on the expression of IL-17 and SOCS1 proteins [131].

Besides their implication in the development of scleroderma and SSc, miRNAs are promising tools for differential diagnosis and assessment of disease progression. According to ROC curve analysis, expression level of miR-138 in fresh whole blood specimens effectively differentiates SSc patients from healthy controls [134]. While significant down-regulation of miR-27a was detected in SSc patients compared to healthy individuals, there was no remarkable difference in the expression of miR-27a between limited and diffused SSc patients [133]. Circulating miR-21-5p, miR-29a-3p, miR-181b-5p, miR-210-3p and miR-223-3p have been established as potential prognostic biomarkers in patients with localized scleroderma [136]. miR-155 and miR-143 content of lung tissue specimens obtained from patients with SSc-associated interstitial lung disease strongly correlated with the progression confirmed by high-resolution computed tomography (HRCT) [135].

miR-214 molecules, delivered by bone marrow MSC-derived exosomes, alleviated skin fibrosis in SSc by the inhibition of IL-33/ST2 axis [138]. Based on its negative regulatory effect on secreted phosphoprotein 1 (SPP1) and ERK signals, miR-27a mimics are potential therapeutic agents for counteracting pathological fibrosis and delaying the progression of SSc [133,137].

### 5.5. Sjögren’s Syndrome

Down-regulation of miR-181d-5p in labial salivary glands of patients with Sjögren’s syndrome contributes to the pro-inflammatory environment by the deregulation of its target TNFα [139]. Enhanced expression of miR-223-3p in ocular wash samples aggravates ocular inflammation through the negative modulation of its direct downstream gene inositol 1,4,5-trisphosphate receptor type 3 (ITPR3) [140]. Important regulators of cellular proteostasis, IFNγ-dependent miR-424-5p and miR-513c-3p were found to be down-regulated and overexpressed in salivary glands from patients, respectively [143]. Expression of B-cell activating factor (BAFF) in salivary gland epithelial cells was found to be regulated by the miR-30 family, the down-regulation of which has been confirmed in primary Sjögren’s syndrome [141].

Combined evaluation of exosome-derived miR-1290 and let-7b-5p in mouth-rinse samples distinguished patients with Sjögren’s syndrome from healthy volunteers with 91.7% sensitivity and 83.3% specificity [144]. According to results obtained from a murine model of Sjögren’s syndrome, miR-181b-5p, miR-322-5p and miR-503-5p may be applied as novel biomarkers of autoimmune dacryoadenitis [142]. In minor salivary gland tissue samples of patients with primary Sjögren’s syndrome, the expression level of miR-92a inversely correlated with disease severity [145].

Attenuation of up-regulated miR-16, miR-142 and miR-223 was found to alleviate heightened immune response and relieve symptoms of patients with Sjögren’s syndrome [147]. Transfection of miR-125b inhibitors into aging MSCs resulted in the promotion of M2 macrophage polarization and inhibition of Th17 differentiation in the spleen [146]. Let-7 family of miRNAs delivered by EVs from induced pluripotent stem cells (iPSC) suppressed the expression of NFκB and TLR4 in a murine model of Sjögren’s syndrome [148].

### 5.6. Autoimmune Vasculitis

Within this heterogeneous group of autoimmune diseases, emerging evidence supports the potential of miRNAs as both biomarkers and novel therapeutic targets.

Since its intercellular transfer via exosomes promotes the development of acute endothelial injury, targeting miR-1287-5p is a promising treatment approach for vascular inflammatory injury induced by microscopic polyangiitis (MPA) [156]. High levels of miR-30a-5p, miR-99a-5p, miR-106b-5p and miR-182-5p in urinary EVs were identified as novel biomarkers of antineutrophilic cytoplasmic antibody (ANCA)-associated vasculitis [153]. Expression of miR-223-3p in circulating EVs discriminates between granulomatosis with polyangiitis (GPA) patients and healthy controls, whereas miR-664-3p was identified as a novel biomarker for disease activity [154]. Expression levels of miR-146a-5p were particularly high in histologically positive temporal artery biopsies of giant cell arteritis (GCA) patients, while levels of miR-424-5p were associated with vision disturbances [149]. In vasculitic skin lesions of adult patients with IgA vasculitis, altered levels of let-7b, miR-148b-3p, miR-155-5p and miR-223-3p were found, among which the expression of miR-223-3p correlated with the severity of purpuras [155]. On the other hand, up-regulation of plasma miR-33 and miR-34 has been confirmed in active periods of childhood IgA vasculitis, as compared to the inactive periods and control groups [151]. A high expression level of circulating miR-24-3p was identified as a potential diagnostic biomarker of Kawasaki disease (KD) [150], whereas evaluation of miR-133a expression can support the distinction between acute and convalescent KD patients [152].

## 6. Neuro-Immunology and miRNAs

### 6.1. Multiple Sclerosis

The absence of homeostatic monocyte control in multiple sclerosis (MS) is facilitated by the deregulation of miRNAs such as miR-124 [161]. miR-142-3p was confirmed to be a critical modulator of both TNF-induced neuronal toxicity [162] and IL-1β-mediated synaptic dysfunction [163]. Levels of miR-150 and miR-181c in cerebrospinal fluid (CSF) were associated with an earlier conversion of clinically isolated syndrome (CIS) to MS [164,165], while the expression of miR-106a-5p in CSF was identified as a marker of early relapsing-remitting MS [166]. 

In MS patients treated with glatiramer acetate for 2 years, serum miR-138-5p levels were down-regulated in cases of no evidence of disease activity (NEDA-3), whereas increased levels of miR-126-3p and miR-146a-5p were measured if confirmed disability progression (CDP) developed [167]. Circulating miR-29b-5p and miR-185-5p can be applied as biomarkers of treatment response to IFN-β [168,169].

### 6.2. Myasthenia Gravis

miRNA fingerprints of PBMCs derived from patients with myasthenia gravis (MG) are featured by the up-regulation of miR-21-5p and miR-106b-3p, and the down-regulation of miR-15b, miR-16 and miR-20b [170]. Dysregulation of let-7c expression was associated with both the initiation and progression of MG through its regulation of IL-10 [171]. Expression of miR-320d, miR-3614-3p and miR-4712-3p in serum exosomes were identified as novel biomarkers of early-onset ocular MG [172]. Responsiveness to immunosuppressive treatment is associated with a recently established miRNA signature composed of miR-181d-5p, miR-323b-3p, miR-340-3p, miR-409-3p and miR-485-3p [173]. Administration of lentiviral-miR-145 resulted in decreased serum anti-AChR IgG levels in a rat model of experimental autoimmune MG [174].

## 7. Implications of miRNAs in Anticancer Immunity

Based on their essential roles in fine-tuning the anticancer properties of immune cells, deregulation of miRNAs was proven to contribute to the development of tumor-promoting microenvironments in a wide range of malignant diseases. EVs derived from M1 macrophages were confirmed to suppress tumor growth, in which the antitumor activities of miR-29a-3p play an important role [175]. Up-regulation of miR-21, miR-28, miR-222 and miR-301a in DCs exposed to tumor antigens promotes the creation of an immunosuppressive tumor microenvironment (TME) [42]. NK cell immune checkpoints are directly regulated by miRNAs, including miR-28, miR-138 and miR-4717 that target programmed cell death protein (PD)-1 and induce T-cell exhaustion [176]. Antitumor CD8 T-cell responses are enhanced by let-7 by promoting memory and antagonizing terminal differentiation [177].

### 7.1. Hematological Malignancies

Enhanced expression of miR-24-3p in exosomes released by acute myeloid leukemia (AML) blasts (R-EXOs) results in increased apoptosis of major CD4^+^ and CD8^+^ T lymphocyte subtypes but the promotion of Treg development [178]. Up-regulation of miR-29b in NK cells was associated with the miR-mediated evasion of NK cell surveillance as an additional mechanism of immune escape in leukemia [179]. Exosomes released in T-cell leukemia are implicated in the suppression of specific immune cells via the impairment of Treg and NK cell functions by miR-150 and miR-181 [180]. Overexpression of miR-181b in B-cells of chronic lymphocytic leukemia (CLL) resulted in an improved antitumor response of cytotoxic T-cells [181]. Overexpression of miR-195 in a human diffuse large B-cell lymphoma (DLBCL) cell line attenuated the immune escape of transformed cells by targeting PD-L1 [182]. Down-regulation of miR-21, miR-130 and miR-155 resulted in decreased growth of cutaneous T-cell lymphoma (CTCL) cell lines, and facilitated CD8^+^ T-cell-mediated cytotoxic activity [183].

### 7.2. Solid Tumors

Production of the neutrophil chemoattractant, C-X-C motif chemokine ligand 1 is counteracted by miR-146a in high-grade serous ovarian cancer, contributing to improved infiltration of the tumor by cytotoxic T lymphocytes [184]. On the other hand, progress of ovarian cancer was promoted by hypoxic tumor-derived exosomal miR-1225-5p that regulates TLR2 and M2 macrophage polarization [185]. B-cell lymphoma 2 (BCL2)-mediated resistance to cytotoxic T lymphocytes is facilitated by miR-192-5p in esophageal squamous cell carcinoma [186]. Immune escape of pancreatic cancer cells is mediated by the up-regulation of miR-1275 that inhibits the expression of axis inhibition protein (AXIN2) in NK cells [187]. Tumor growth in pancreatic cancer is also counteracted by NK cells through the exosomal transfer of let-7b-5p [188]. In a murine model of hepatocellular carcinoma (HCC), antitumor immunity was enhanced by miR-206 through the disruption of communication between T regs and malignant hepatocytes [189]. In nonsmall-cell lung carcinoma (NSCLC), improved anticancer immunity was associated with the administration of miR-4458 mimics and consequent down-regulation of PD-L1 [190]. Tumor immune escape of papillary thyroid carcinoma cells in distant organs was found to be aided by exosomal miR-519e-5p, through the inhibition of Notch signaling [191]. In bladder cancer, tumor immune escape was promoted by the down-regulation of miR-27a, miR-30c and miR-135a [192]. In a murine model of melanoma, antitumor properties of DCs were impaired by miR-22 [193].

## 8. Applications of miRNAs in Transplantation Immunology

Post-transplant events including disease recurrence and allograft rejection are of key significance with regard to patient outcome, in which miRNAs play an essential role [194]. Growing numbers of miRNAs have been proposed as novel biomarkers for allograft status including early detection of rejection [195,196].

miRNA fingerprints identified in kidney allograft biopsy specimens represent an additional diagnostic tool to differentiate between antibody-mediated rejection, T-cell-mediated rejection, acute tubular necrosis and BK polyomavirus-associated nephropathy [197]. Based on a five-element miRNA panel composed of circulating miR-15b, miR-16, miR-103a and miR-106amiR-107, T-cell-mediated vascular rejection can be distinguished from stable graft function [198]. Evaluation of circulating miR-15b, miR-103a and miR-106a expression discriminated patients with stable function of kidney graft from those with urinary tract infection and T-cell mediated rejection [199]. Urinary exosomal miRNA profile including miR-21-5p, miR-31-5p and miR-4532 can be applied as a noninvasive biomarker of acute rejection in kidney transplant recipients [200].

miR-122, miR-146a, miR-155-5p, miR-181a-5p and miR-192 were identified as potential biomarkers of acute rejection following liver transplantation [201,202]. miR-155-5p and miR-181a-5p are also involved by a recently published logistic model for rejection prediction and diagnosis, according to which liver graft dysfunction can be earlier identified, contributing to a more efficient adjustment of immunosuppressive therapy [203].

Prolonged survival of rat cardiac allografts following the intrathymic and intravenous injection of MSCs was associated with the down-regulation of miR-155 expression [204]. Production of IL-10 by peripheral B-cells was suppressed by miR-98 following cardiac transplantation, that was successfully inhibited by the administration of anti-miR-98, resulting in improved survival of mice received allografts [205]. Combined evaluation of circulating miR-144-3p and miR-652-3p can be applied as biomarkers for acute cellular rejection in heart transplanted patients [206]. On the other hand, no clinical utility of circulating miR-10a and miR-92a have been confirmed in heart allograft recipients [207].

Specific plasma miRNA fingerprints, consisting of miR-30a, miR-93*, miR-155, miR-181a, miR-199a-3p, miR-377 and miR-423, may serve as independent biomarkers for the early detection and prognosis prediction of acute graft-versus-host-disease (GVHD) following hematopoietic stem cell transplantation (HSCT) [208,209]. According to a murine model of acute GVHD, the infusion of miR-223 agomiR resulted in symptom relief and decreased production of inflammatory cytokines [210]. Chronic GVHD has been associated with the dysregulated expression of miR-122-5p, miR-148-3p, miR-192-5p, miR-365-3p and miR-378-3p [211,212].

Recent great development of cellular therapy is well characterized by the today’s commercially available chimeric antigen receptor (CAR) T-cells [213]. Up-regulation of miR-146a and the miR-29 family have been implicated in improved antitumor effects of CAR-T-cell therapy through the metabolic shift of T-cells toward oxidative phosphorylation and fatty acid oxidation [214]. Increased anticancer functions of such anti-CD19 CAR-T-cells have been registered, in which miR-155 was overexpressed by a retroviral vector construct [215]. Increased expression of miR-148a-3p and miR-375 were associated with good response to CAR-T-cell therapy in B-cell acute lymphoblastic leukemia [216].

## 9. Impact of Immunomodulation on miRNA Fingerprints

A growing body of evidence supports the remarkable influence of immunomodulatory interventions on miRNA fingerprints. These alterations provide novel translational and clinical applications and facilitate advances in prevention activities, aimed at the personalized management of immune dysregulation.

### 9.1. Pharmacological Interventions

The expression of miR-17-92 polycistron and the key miRNA processing enzymes, Drosha and Dicer were found to be significantly down-regulated by dexamethasone treatment, contributing to the glucocorticoid-induced apoptosis of lymphocytes [217]. Besides the impact of glucocorticoids on the transcription of miRNAs, the expression and function of glucocorticoid receptors are also regulated by miRNAs [2,218]. Increased apoptosis of plasmacytoid DCs, facilitated by miR-29b and miR-29c, may improve the efficacy of glucocorticoid treatment in SLE [219]. Methotrextae (MTX) treatment in early RA resulted in differential expression patterns of numerous serum EV-derived miRNAs, including miR-212-3p, miR-338-5p, miR-410-3p and miR-537, that regulate the pathogenicity of synovial fibroblasts [220]. Up-regulation of miR-877-3p in response to MTX in RA-FLS was confirmed to attenuated both the proliferation and cytokine production of synoviocytes [221]. Differentiation and IL-17 production of Th17 cells was inhibited by hydroxychloroquine (HCQ) treatment in SLE patients, that was associated with increased expression of miR-590 [222].

### 9.2. Biologicals

Combination treatment with anti-TNFα and nonbiologic disease-modifying antirheumatic drugs (DMARDs) resulted in the significant up-regulation of miR-16-5p, miR-23-3p, miR125b-5p, miR-126-3p, miRN-146a-5p and miR-223-3p in RA patients [110]. Tocilizumab has been confirmed to decrease angiogenesis and the production of matrix metalloproteinases (MMPs) in RA-FLS through the down-regulation of miR-146a-5p and miR-214, respectively [223,224]. In RA patients treated with adalimumab and MTX, remitting and nonremitting cases could be distinguished at 3 months based on the expression changes in miR-27a-3p, a novel biomarker of successful remission induction [225]. Administration of low-dose rituximab therapy resulted in decreased expression levels of serum exosomal miR-150-5p in AChR-positive refractory myasthenia gravis patients [226].

### 9.3. Physical Exercise

miRNAs play a remarkable role in exercise-induced effects on both innate and adaptive immunity by affecting the distribution and cytokine production of immune cells. miR-15, miR-29c, miR-30a, miR-142/3, miR-181a and miR-338 have been confirmed to mediate such immunomodulatory effects of acute bouts of endurance training [227]. Age-related defects in adaptive immunity due to the decline of miR-181a expression in T-cells was found to be ameliorated by regular aerobic exercise [227]. Beneficial effects of high-intensity interval training (HIIT) in autoimmune diseases such as reduced disease activity of RA and vasculoprotective adaptations to physical exercise have been associated with the up-regulation of miR-24, miR-96-5p, miR-133a, miR-133b and miR-143 [228,229,230].

### 9.4. Metabolism and Nutrition

Overexpression of miR-99a, that is frequently down-regulated in RA and SLE, mitigated the development of experimental autoimmune encephalomyelitis (EAE) in mice through the inhibition of mammalian target of rapamycin (mTOR)-dependent glycolysis [231]. Targeting glutamine metabolism is a potential treatment approach in SLE to inhibit plasmablast differentiation [232] and in cancer to overcome tumor immune evasion as well [233]. Glutaminase enzyme is repressed by miR-137, miR-153 and miR-200a-3p, to which miRNAs can therefore be applied as novel therapeutic targets [234,235,236]. Vitamin D has a suppressive role on autoimmunity by promoting the differentiation of Tregs and reducing the secretion of inflammatory cytokines [237]. Attenuation of TLR-mediated inflammation by vitamin D receptor (VDR) signaling is mediated by the down-regulation of miR-155 and the subsequent stimulation of SOCS1 production [238]. In a murine model of EAE, the protective roles of vitamins A and D have been associated with the up-regulation of GATA3 and miR-27-3p [239]. In elderly Swedish citizens, combined supplementation of selenium and coenzyme Q10 (CoQ_10_) resulted in reduced levels of markers of inflammation and significant changes in the expression of more than 100 different miRNAs [240].

## 10. Discussion

Advances in diagnostics and novel therapeutic options including biologicals contributed to the improved treatment outcomes and life quality of patients with autoimmune diseases. However, the remarkable interindividual heterogeneity in both symptoms and responsiveness to therapy necessitates further research aiming the personalized management of immune dysregulation. Cornerstones of the further achievements include the establishment of noninvasive biomarkers for differential diagnosis, prediction of prognosis and treatment response. Identification of novel therapeutic targets can facilitate new, less toxic therapeutic approaches with a remarkable reduction in side effects.

The clinically significant effect of epigenetic regulatory mechanisms on the course of systemic autoimmune diseases is well indicated by their widespread interactions with the cells of innate and adaptive immunity [241]. First definition of the term epigenetics was established by Conrad Hal Waddington as the “causal interactions between genes and their products, which bring the phenotype into being” in 1942 [242]. DNA methylation, histone modification, chromosome remodeling and noncoding RNA molecules including miRNAs are responsible for the fine-tuning of gene expression patterns, aiming the efficient adaptation to changing environmental influences. Therefore, reversibility and a remarkable degree of plasticity are fundamental hallmarks of epigenetic alterations.

miRNAs are approximately 20-nucleotide-long, evolutionarily conserved noncoding RNA molecules, providing a wide range of translational and clinical applications in diseases featured by immune dysregulation [243]. Besides the fine-tuning of differentiation and function of immune cells, miRNAs are key players of anticancer immunity and are also implicated in transplantation immunology. miRNA fingerprints have been confirmed to undergo reversible alterations in case of infections, immunodeficiencies and autoimmune diseases.

The growing number of miRNA biomarkers in serum or urine can enable a more advanced diagnostic and prognostic classification, with more accurate prediction of long-term outcome and treatment response [159]. Patient selection and response prediction to biological therapy is one of the most promising fields of application [244]. Since miRNA fingerprints change during the course and treatment of autoimmune diseases, miRNAs can also be suitable for monitoring disease progression and treatment response [109].

Replacement or inhibition of miRNAs has enormous therapeutic potential, with constantly expanding possibilities [245]. Single-stranded oligonucleotides as therapeutic agents were first proposed in the late 1970s [246]. The first antisense oligonucleotide (ASO)-based drug to be approved was fomivirsen in 1998 for the treatment of CMV retinitis in immunocompromised patients [247]. Since then, small interfering RNAs (siRNAs) such as givosiran, inclisiran and patisiran have been approved for the treatment of rare metabolic disorders, and further modes of actions have been established such as locked nucleic acid (LNA)-based oligonucleotides [248]. LNA miRNA inhibitors have the 2′-O and 4′-C atoms of the ribose ring hooked up through a methylene link, resulting in increased stability against degradation [249].

Using the technologies mentioned above, inhibition of the miR-17-92 cluster might be useful in autoimmune diseases driven by follicular Th cells [250]. Induction of miR-146a would have immunosuppressive effects by enhancing the function of Treg cells [251], while the administration of miR-210 mimics might dampen Th17-driven inflammation [252]. It is important to note, that epigenetic agents also modify miRNA fingerprints. Hypomethylating agents restored the expression of miRNAs that are repressed by promoter hypermethylation [253,254]. Inhibition of histone deacetylase (HDAC) enzymes induced significant changes in the expression of >60 different miRNA species in breast cancer cell lines [255].

Besides the pharmacological interventions for immunosuppression, different forms of nonpharmacological immunomodulation also exert considerable influence on the expression of miRNAs, such as the exercise-induced changes in miRNA patterns are involved in the regulation of cytokine production and the entire inflammatory process [227]. More detailed knowledge of the molecular effects of physical exercise and nutrition on the immune system may facilitate self-tailored lifestyle recommendations and advances in prevention.

Despite the development of molecular biological techniques and expanding opportunities for the modulation of miRNA expression levels, important challenges remain to be overcome.

Standardization of the methodology for the evaluation of miRNA expression fingerprints is of key significance (Figure 2). Comparability of results requires standardized preanalytic conditions including sampling sources and circumstances of storage. miRNA fingerprints are most commonly evaluated from serum or whole blood specimens; however, bone marrow, saliva, urine, synovial fluid, tear, bile, cerebrospinal fluid (CSF), breast milk, ascitic fluid, cartilage, tissue homogenizates from biopsies and formalin-fixed paraffin-embedded tissue samples are also widely used. The choice between samples therefore depends on the question to be investigated and ethical issues, using the least invasive sampling procedure possible.

Isolation protocols of miRNA fraction and genes for normalization also should be uniformly established, since different data normalization processes may result in several inconsistency and irreproducibility [249]. Trizol/TRI-Reagent based isolation of total RNA including miRNA fraction is regarded as a robust reproducible method, without any tendency of the obtained miRNA samples to degradation when stored properly [256]. On the other hand, recent reports on methodology of miRNA extraction demonstrated higher efficiency of column-based isolation kits compared to Trizol [257,258]. However, different kits apply nonidentical protocols, resulting in variable RNA yield and purity [259,260].

Though methodologies for miRNA microarray and noncoding RNA sequencing have already been established, reverse transcription followed by quantitative polymerase chain reaction (RT-qPCR) is the preferred choice to measure miRNA expression levels in biofluids, as other methods are currently less sensitive [261]. Difficulties in determining the normal ranges of miRNA expression levels and age-associated differences in miRNA fingerprints among healthy individuals should also be mentioned.

Since a single miRNA can target multiple mRNAs simultaneously, and one mRNA can also be regulated by several different miRNAs [262], the therapeutic modulation of miRNA levels may have pleiotropic effects depending on the microenvironment. Context-specific functions of miRNAs are also influenced by numerous protein–protein and protein–RNA interactions [263]. Uptake of miRNA mimics and inhibitors by different organs may result in unpredictable side effects [249]. However, vector tropism restriction by tissue-specific miRNAs can be adapted to maximize specificity without compromising efficacy [264].

Successful administration of miRNA replacements and antagonists requires the development of efficient and safe delivery systems. Though the substitution of down-regulated miRNAs is a promising treatment approach, earlier attempts for the delivery of synthetic oligonucleotides with 2′-alkylated nucleotides or phosphorothioate backbone modifications could be performed only at the expense of pharmakokinetic properties and stability [265]. Using a polyurethane–polyethylene imine copolymer (PU-PEI), intracranially delivered miR-145 exerted synergistic effects in conjunction with chemotherapy and radiotherapy in glioblastoma multiforme [266]. A recent report on methodology by Kasina et al. describes a miRNA delivery system with cationic polylactic-co-glycolic acid (PLGA)-poly-L-histidine nanoparticles, the efficacy of which has been confirmed in vivo [267]. For the inhibition of distinct miRNAs, also several methods have been developed with special technical challenges. Based on the lack of tolerance of the RNA interference mechanism for chemical modifications, conjugation to or packaging in delivery systems seems to be required [268]. A covalent conjugate with N-acetlygalactosamine (GalNAc) has been developed for the delivery of anti-miR-122, from which compound (RG-101) has been shown to be effective and safe in preclinical animal models [269].

During the previous two decades, growing numbers of miRNA and anti-miRNA delivery vectors have been developed [270]. Among viral based delivery vectors, retro- and lentiviral vectors are featured by stable transgene expression but also associated with the risk of insertional mutagenesis [271]. Though bacteriophage-based virus-like particle (VLP) vectors are associated with low carcinogenic potential, their loading capacity is low [272]. Advantages of adeno-associated vectors include high transduction efficiency over a wide variety of cells, but their packaging capacity is low [273]. Challenges featuring nonviral-based delivery vectors include low efficiency and cytotoxicity in case of polymeric and lipid-based systems [274]. Extracellular vesicle-based delivery is a remarkably promising strategy due to its high packaging capacity, low immunogenicity and tissue-specific delivery; however, it is necessary to develop a feasible method for its large-scale production [270].

## 11. Concluding Remarks

Further improvement of treatment results and the life-quality of patients with immune dysregulation requires personalized management of diseases, based on novel biomarkers and therapeutic targets. The key role of miRNAs in the regulation of immune response has been proven both in health and diseases. The remarkable interindividual differences in both symptoms and treatment response featuring autoimmune diseases are in accordance with the plasticity and widespread interacting networks of epigenetic regulatory mechanisms, that provide promising opportunities for beneficial adaptation to environmental factors, including lifestyle interventions. Although important challenges still await solutions, clinical applications of miRNAs may contribute to less-toxic and increasingly targeted treatment approaches to diseases of the immune system.

## Figures and Tables

**Figure 1 ijms-25-01942-f001:**
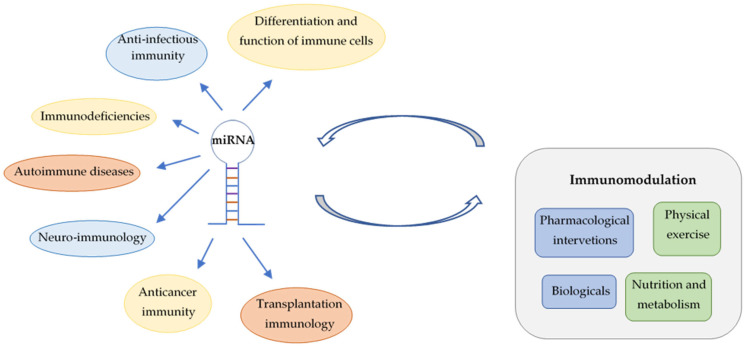
Involvement of miRNAs in immune regulation and interactions between miRNAs and different types of immunomodulation.

**Figure 2 ijms-25-01942-f002:**
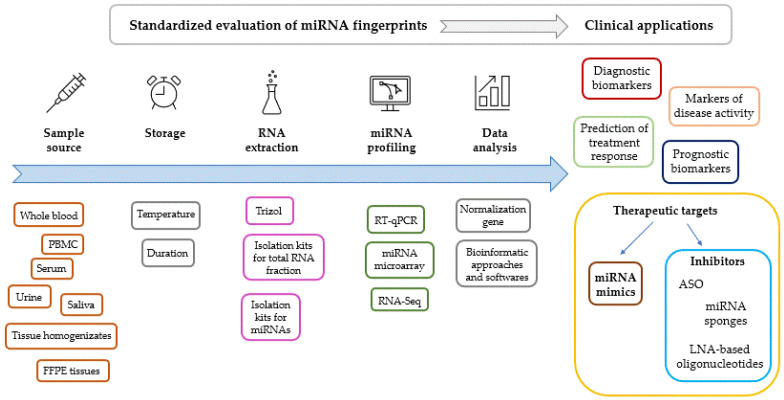
Factors to be standardized during the evaluation of miRNA expression levels. Clinical applications of miRNA fingerprints. Abbreviations: ASO: antisense oligonucleotide, FFPE: formalin-fixed, paraffin-embedded tissue, LNA: locked nucleic acid, PBMC: peripheral blood mononuclear cells, RT-qPCR: reverse transcription and polymerase chain reaction.

**Table 1 ijms-25-01942-t001:** Involvement of miRNAs in anti-infectious immune responses.

Microorganism	miRNA	Target/Mechanism of Action
**Virus**		
HBV	miR-1231	HBV core mRNA [57]
HCV	miR-122	viral translation [58]
Influenza A (H1N1)	miR-21, miR-99a	pulmonary repair and inflammatory processes [59]
HMPV	miR-4634	IFN production [60]
HSV1	miR-24	STING [61]
EBV	miR-BART17-3p	RIG-I like receptor antiviral pathway [62]
ZIKV	miR-30c, miR-194, miR-302b, miR-302c	microcephaly and brain lesions [63]
HIV-1	miR-155, miR-181-5p	TLR, DDX3X [64]
SARS-CoV2	let-7, miR-17-5p, miR-18a-5p, miR-19a-3p, miR-20a-5p, miR-124, miR-155, miR-200c-3p, miR-223	production of inflammatory cytokines [65,66], development of fibrotic lung lesions [67]
**Bacteria**		
*M. tuberculosis*	miR-155, miR-215-5p	fusion of autophagosomes with lysosomes [68,69]
* M. avium * subsp. *paratuberculosis*	miR-150	PDCD4 [70]
*S. pneumoniae*	miR-223	pulmonary inflammation [71]
*N. meningitidis*	miR-146a	induction of endotoxin tolerance [72]
*L. pneumophila*	miR-125a-5p	NTAN1 [73]
*K. pneumoniae*	miR-181a-5p	alleviation of lung injury [74]
*B. abortus*	miR-125b-5p	TNFα production [75]
**Fungi**		
*C. albicans*	miR-16-1, miR-17-3p, miR-155	SOCS1 [76,77]
*C. glabrata*	miR-146a	production of inflammatory cytokines [78]
*A. fumigatus*	miR-21-5p, miR-26-5p, miR-142-3p, miR-142-5p	markers of invasive aspergillosis [79]
**Parasite**		
*B. malayi*	miR-34	CXCL10/CXCL11/CXCR3 [80]
*T. cruzi*	miR-155	TNFα and IFN production, NK cell function [81]
*L. donovani*	miR-21, miR-146b-5p	increased expression of IL-6 and STAT3 [82]
*S. japonicum*	miR-146b, miR-155	hepatic inflammation [83]
*P. falciparum*	miR-27a, miR-150, let-7i	markers of cerebral malaria [84]

**Table 2 ijms-25-01942-t002:** Implication of miRNAs in autoimmune diseases. Roles in pathogenesis and clinical applications.

Disease	Pathogenesis	Diagnosis/Disease Activity	Prognosis/Treatment Response	Therapy
**Rheumatoid arthritis**	miR-146a (rs2431697) [105], miR-200a-3p [106]	miR-21-5p [107], miR-25, miR-371b, miR-378d, miR-483, miR-642b [108]	miR-19b [109], miR-23, miR-223 [110]	miR-155 [111], miR-340-5p [112]
**Systemic lupus erythematosus**	miR-17-92 (rs4284505) [113], miR-146a [114], miR-365a-3p [115], miR-4689 [116]	miR-21 [117], miR-124-3p, miR-146, miR-320b, miR-377-3p [118,119]	miR-181a, miR-223 [120], miR-485-5p [121]	miR-7 [122], miR-30a [123], miR-181d-5p [124], miR-590-3p [125]
**Antiphospholipid syndrome**	miR-326, miR-483-3p [126]	miR-19b-3p, miR-20a-5p [127]		miR-146a-5p [128]
**Systemic sclerosis, scleroderma**	miR-21 [129], miR-126 [130], miR-155-5p [131]	miR-21 [132], miR-27a [133], miR-29a, miR-138 [134]	miR-21-5p, miR-29a-3p, miR-143, miR-155 [135], miR-181b-5p, miR-210-3p, miR-223-3p [136]	miR-27a [137], miR-214 [138]
**Sjögren’s syndrome**	miR-181d-5p [139], miR-223-3p [140]	miR-30 [141], miR-181b-5p, miR-322-5p, miR-424-5p, miR-503-5p [142], miR-513c-3p [143], miR-1290, let-7b-5p [144]	miR-92a [145]	miR-16, miR-125b [146], miR-142, miR-223 [147], let-7 [148]
**Autoimmune vasculitis**	miR-424-5p [149]	miR-24-3p [150], miR-30a-5p, miR-33, miR-34 [151], miR-99a-5p, miR-106b-5p, miR-133a [152], miR-146a-5p [149], miR-148b-3p, miR-155-5p, miR-182-5p [153], miR-223-3p, miR-664-3p [154], let-7b [155]		miR-1287-5p [156]

## Data Availability

No new data were created or analyzed in this study. Data sharing is not applicable to this article.

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
