# Peer review of "Role of microRNAs in Immune Regulation with Translational and Clinical Applications"

_ijms, 2024, doi:10.3390/ijms25031942_

Round 1

Reviewer 1 Report

Comments and Suggestions for Authors

I thank Zsuzsanna Gaál for this interesting review. This review is important because the results of miRNA studies have not been previously systematically compiled. This review is quite detailed and examines the role of various miRNAs in a number of diseases. Major focus is put on the translational clinical applications of miRNAs. Understanding the role of microRNAs has therapeutic potential.

However, there are three notes:

1. Figure 1. Please use a higher quality image.

2. Please expand section 3.1. During viral infections, there are more microRNAs described in the literature. For example, in COVID-19 the following roles have been described for: miR-125b-5p, miR-155-5p, let7b-5p, miR-148a, miR-31, miR-29, miR-126, miR-17 and others.

3. The authors mention the need for standardization of the methodology for the evaluation of miRNA. Please add a description of the methodologies used.

Author Response

Thank you very much for reviewing this manuscript about the role and clinical significance of miRNAs in disorders of immune regulation.

Please find my answears to the comments and recommendations below.

  1. Quality image of Figure 1 (page 3) has been improved. Figure 2 (page 26) has been completed with clinical applications of miRNAs.

  1. Section 3.1. has been expanded. In the first paragraph, significance of miR-155 is included in the revised version of the manuscript, as a „conductor miRNA” in immune regulation (lines 197-201). The role of miR-24 in evading cellular antiviral response against HSV1 is also included (lines 212-214). The description of the role of miRNAs in COVID-19 is more detailed in the revised version. Potential miRNA biomarkers (such as miR-21-5p, miR-22-3p, miR-155-5p and miR-224-5p) are listed (lines 228-234) according to recent reports, and the pathogenetic role of deregulated anti-fibrotic miRNAs (including miR-17-5p, miR-19a-3p, miR-20a-5p and miR-200c-3p) in the development of fibrotic lung lesions in post-COVID patients is also included (lines 236-238).

Table 1 (involvement of miRNAs in anti-infectious immune responses) has been completed according to the novel citations included in section 3.1. (page 7).

  1. In the revised version of the manuscript, a detailed description is included about the challenges of clinical applications of miRNAs, including the necessity of standardizing methodologies. Different sample sources (lines 720-725), methods for RNA extraction (lines 728-733) and miRNA profiling including RT-qPCR (lines 742-745) are also included. The description of standardized methodology is included in the Discussion as follows (lines 717-747).

„Standardization of the methodology for the evaluation of miRNA expression fingerprints is of key significance (Figure 2). Comparability of results requires standardized preanalytic conditions including sampling sources and circumstances of storage. miRNA fingerprints are most commonly evaluated from serum or whole blood specimens, however, bone marrow, saliva, urine, synovial fluid, tear, bile, cerebrospinal fluid (CSF), breast milk, ascitic fluid, cartilage, tissue homogenizates from biopsies and formalin-fixed paraffin-embedded tissue samples are also widely used. The choice between samples therefore depends on the question to be investigated and ethical issues, using the least invasive sampling procedure possible.

Isolation protocols of miRNA fraction and genes for normalization also should be uniformly established, since different data normalization processes may result in several inconsistency and irreproducibility [249]. Trizol/TRI-Reagent based isolation of total RNA including miRNA fraction is regarded as a robust reproducible method, without any tendency of the obtained miRNA samples to degradation when stored properly [256]. On the other hand, recent reports on methodology of miRNA extraction demonstrated higher efficiency of column-based isolation kits compared to Trizol [257, 258]. However, different kits apply non-identical protocols, resulting in variable RNA yield and purity [259, 260].

Though methodologies for miRNA microarray and non-coding RNA sequencing have already been established, reverse transcription followed by quantitative polymerase chain reaction (RT-qPCR) is the preferred choice to measure miRNA expression levels in biofluids, as other methods are currently less sensitive [261]. Difficulties in determining the normal ranges of miRNA expression levels and age-associated differences in miRNA fingerprints among healthy individuals should also be mentioned.”

Please note that numbering of citations has changed due to the novel citations inserted. Citations are also included within Table 1 and Table 2 in the revised version.

Title of the manuscript has also been modified, which is labeled with yellow colour, similarly to all other modifications.

Reviewer 2 Report

Comments and Suggestions for Authors

The review article by Zsuzsanna Gaál discusses current achievements in the field of microRNAs in immunity. It presents a comprehensive overview of the topic and is very well written. The article might be interesting for a wide audience of readers. Some minor corrections are needed before publishing.

  1. The title is sub-optimal. "Clinical applications" is somehow linked to involvement, which is grammatically wrong.
  2. In Chapter 2, if this chapter aims to describe all immune cells, why not include a sub-chapter on granulocytes, mast cells, or title it "other immune cells" to cover all immune cell types? Otherwise, it is not understandable why only these selected immune cells are presented.
  3. In Chapter 3, "Anti-infectious immunity," I found that the subchapters "virus," "bacteria," etc., would benefit from the extended title, for example, anti-viral immunity or a similar title. Otherwise, it looks like miRNAs are studied in viruses or bacteria.
  4. References to each data in Table 1 and Table 2 would be good to add.
  5. Figure 2 requires improvement. Enlarge text boxes and improve the quality of text. Add a section on clinical applications of miRNA targeting strategies or treatments based on miRNAs expressions, especially because "clinical implications" are included in the title.
  6. The discussion contains a paragraph on challenges and therapeutical drugs. I would suggest discussing open questions on how to deliver nucleotide-based drugs targeting miRNAs.

Author Response

Thank you very much for reviewing this manuscript about the role and clinical significance of miRNAs in disorders of immune regulation.

Please find my answears to the comments and recommendations below.

  1. Title has been modified as follows: „Role of microRNAs in immune regulation with translational and clinical applications”. Modified title is labeled with yellow in the revised version of the manuscript, similarly to all other modifications.

  1. Chapter 2 has been completed with the description of miRNAs that regulate neutrophil granulocytes and mast cells (lines 164-187):

2.6. Neutrophil

miRNAs play a fundamental role in both differentiation and function of neutrophil granulocytes. While the elevated expression level of miR-155 (targeting PU.1 transcription factor) features myeloblasts and promyelocytes, high levels of miR-27a (down-regulating RUNX1 transcription factor) are characteristic to myelocyte stages, and miR-223 is the most well-known miRNA that facilitates the process of granulopoiesis [45]. Regulation of neutrophil actin cytoskeleton by miR-142 is essential for protection against bacterial infections at skin wound sites [46]. Based on the results of experimental inhibition, miR-328 and miR-183/96/182 cluster influence the phagocytic capacity of neutrophil granulocytes [47, 48], while zebrafish miR-722 was identified as a suppressor of chemotaxis in human neutrophil like cells [49].

2.7. Mast cell

Mast cells are key regulators of allergic disorders and immediate hypersensitivity reactions [50]. Compared to other hematopoietic lineages, bone marrow-derived mast cells expressed lower levels of miR-223, and higher levels of miR-24, miR-26a and miR-27a [51]. Dinitrophenol (DNP)-stimulated and resting mast cells also showed different miRNA fingerprints including the expression of let-7i, that was confirmed to inhibit the process of degranulation by targeting exocyst complex component 8 (Exco8) [52]. Degranulation is also regulated by miR-21 through the inhibition of p38 pathway [53]. According to recently published data, exosomal miRNAs derived from mast cells are implicated in the development of tumor metastasis, such as the transfer of miR-490 to hepatocellular carcinoma cells resulted in the inhibition of metastasis formation through the regulation of EGFR/AKT/ERK pathway [54].”

  1. Titles of subchapters have been extended within Chapter 3 as follows:

3.1. Antiviral immunity

3.2. Immune response against bacteria

3.3. miRNA fingerprints in fungal infections

3.4. Anti-parasite immunity

3.5. miRNA biomarkers in sepsis

In the first paragraph of subchapter 3.1., significance of miR-155 is included in the revised version of the manuscript, as a „conductor miRNA” in immune regulation (lines 197-201). The role of miR-24 in evading cellular antiviral response against HSV1 is also included (lines 212-214). The description of the role of miRNAs in COVID-19 is more detailed in the revised version (lines 228-238).

Table 1 (page 7, involvement of miRNAs in anti-infectious immune responses) has been completed according to the novel citations included in section 3.1.

Please note that numbering of citations has changed due to the novel citations inserted.

  1. References are included within Table 1 (page 7) and Table 2 (page 12) in the revised version of the manuscript.

  1. Figure 2 (page 26) was completed with the clinical applications of miRNAs. Newly included abbreviations (ASO: antisense oligonucleotide and LNA: locked nucleic acid) have been inserted into the figure legend. Image quality has been improved with enlarged text boxes as well.

  1. In the revised version of the manuscript, a detailed description is included in the discussion about the challenges of clinical applications of miRNAs, including the necessity of standardizing methodologies (lines 720-747) and delivery vectors too (lines 756-784):

„Successful administration of miRNA replacements and antagonists requires the development of efficient and safe delivery systems. Though the substitution of down-regulated miRNAs is a promising treatment approach, earlier attempts for the delivery of synthetic oligonucleotides with 2’-alkylated nucleotides or phosphorothioate backbone modifications could be performed only at the expense of pharmakokinetic properties and stability [265]. Using a polyurethane-polyethylene imine copolymer (PU-PEI), intracranially delivered miR-145 exerted synergistic effects in conjunction with chemotherapy and radiotherapy in glioblastoma multiforme [266]. A recent report on methodology by Kasina et al describes a miRNA delivery system with cationic polylactic-co-glycolic acid (PLGA)-poly-L-histidine nanoparticles, whiches efficacy has been confirmed in vivo [267]. For the inhibition of distinct miRNAs, also several methods have been developed with special technical challenges. Based on the lack of tolerance of the RNA interference mechanism for chemical modifications, conjugation to or packaging in delivery systems seems to be required [268]. A covalent conjugate with N-acetlygalactosamine (GalNAc) has been developed for the delivery of anti-miR-122, which compound (RG-101) has been shown to be effective and safe in preclinical animal models [269].

During the previous two decades, growing numbers of miRNA and anti-miRNA delivery vectors have been developed [270]. Among viral based delivery vectors, retro- and lentiviral vectors are featured by stable transgene expression but also associated with the risk of insertional mutagenesis [271]. Though bacteriophage-based virus-like particle (VLP) vectors are associated with low carcinogenic potential, their loading capacity is low [272]. Advantages of adeno-associated vectors include high transduction efficiency over a wide variety of cells, but their packaging capacity is low [273]. Challenges featuring non-viral based delivery vectors include low efficiency and cytotoxicity in case of polymeric and lipid-based systems [274]. Extracellular vesicle-based delivery is a remarkably promising strategy due to its high packaging capacity, low immunogenicity and tissue-specific delivery, however, it is necessary to develop a feasible way of large-sclae production [270].

Round 2

Reviewer 1 Report

Comments and Suggestions for Authors

The new additions to the manuscript made a big difference. The quality of the paper had improved, and all my questions were addressed. No more comments.